# Injury-related cell death and proteoglycan loss in articular cartilage: Numerical model combining necrosis, reactive oxygen species, and inflammatory cytokines

**Joonas P. Kosonen**[1]*, **Atte S. A. Eskelinen**[1], **Gustavo A. Orozco**[1,2], **Petteri Nieminen**[3], **Donald D. Anderson**[4], **Alan J. Grodzinsky**[5], **Rami K. Korhonen**[1], **Petri Tanska**[1]

1 Department of Technical Physics, University of Eastern Finland, Kuopio, Finland, 2 Department of Biomedical Engineering, Lund University, Lund, Sweden, 3 Institute of Biomedicine, University of Eastern Finland, Kuopio, Finland, 4 Departments of Orthopedics & Rehabilitation and Biomedical Engineering, University of Iowa, Iowa City, Iowa, United States of America, 5 Departments of Biological Engineering, Electrical Engineering and Computer Science, and Mechanical Engineering, Massachusetts Institute of Technology, Cambridge, Massachusetts, United States of America

* joonas.kosonen@uef.fi

**Data Availability Statement:** All the underlying data are available in Fairdata Research Data

## Abstract

Osteoarthritis (OA) is a common musculoskeletal disease that leads to deterioration of articular cartilage, joint pain, and decreased quality of life. When OA develops after a joint injury, it is designated as post-traumatic OA (PTOA). The etiology of PTOA remains poorly understood, but it is known that proteoglycan (PG) loss, cell dysfunction, and cell death in cartilage are among the first signs of the disease. These processes, influenced by biomechanical and inflammatory stimuli, disturb the normal cell-regulated balance between tissue synthesis and degeneration. Previous computational mechanobiological models have not explicitly incorporated the cell-mediated degradation mechanisms triggered by an injury that eventually can lead to tissue-level compositional changes. Here, we developed a 2-D mechanobiological finite element model to predict necrosis, apoptosis following excessive production of reactive oxygen species (ROS), and inflammatory cytokine (interleukin-1)-driven apoptosis in cartilage explant. The resulting PG loss over 30 days was simulated. Biomechanically triggered PG degeneration, associated with cell necrosis, excessive ROS production, and cell apoptosis, was predicted to be localized near a lesion, while interleukin-1 diffusion-driven PG degeneration was manifested more globally. Interestingly, the model also showed proteolytic activity and PG biosynthesis closer to the levels of healthy tissue when pro-inflammatory cytokines were rapidly inhibited or cleared from the culture medium, leading to partial recovery of PG content. The numerical predictions of cell death and PG loss were supported by previous experimental findings. Furthermore, the simulated ROS and inflammation mechanisms had longer-lasting effects (over 3 days) on the PG content than localized necrosis. The mechanobiological model presented here may serve as a numerical tool for assessing early cartilage degeneration mechanisms and the efficacy of interventions to mitigate PTOA progression.

Storage Service (https://etsin.fairdata.fi/dataset/131ac4ec-ffe3-48b5-8d40-0c0bc079b7d6). DOI: https://doi.org/10.23729/518aa4b9-bb06-45cb-b768-630099c6732c.

**Funding:** We acknowledge funding support from: The Doctoral Programme in Science, Technology and Computing (SCITECO) (JPK), Strategic funding of the University of Eastern Finland (PT), Academy of Finland (grant nos. 334773 – under the frame of ERA PerMed, 324529) (RKK), Novo Nordisk Foundation (grant no. NNF21OC0065373, the Center for Mathematical Modeling of Knee Osteoarthritis (MathKOA)) (RKK), Alfred Kordelin Foundation (grant no 190317) (PT), Maire Lisko Foundation (PT), Sigrid Juselius Foundation (RKK, DDA), Saastamoinen Foundation (PT), Instrumentarium Science Foundation (ASAE), the Swedish Research Council (2019-00953—under the frame of ERA PerMed). The funders had no role in study design, data collection and analysis, decision to publish, or preparation of the manuscript.

**Competing interests:** The authors have declared that no competing interests exist.

## Author summary

Osteoarthritis is one of the most common musculoskeletal diseases. When osteoarthritis develops after a joint injury, it is designated as post-traumatic osteoarthritis. A defining feature of osteoarthritis is degeneration of articular cartilage, which is partly driven by cartilage cells after joint injury, and further accelerated by inflammation. The degeneration triggered by these biomechanical and biochemical mechanisms is currently irreversible. Thus, early prevention/mitigation of disease progression is a key to avoiding PTOA. Prior computational models have been developed to provide insights into the complex mechanisms of cartilage degradation, but they rarely include cell-level cartilage degeneration mechanisms. Here, we present a novel approach to simulate how the early post-traumatic biomechanical and inflammatory effects on cartilage cells eventually influence tissue composition. Our model includes the key regulators of early post-traumatic osteoarthritis: chondral lesions, cell death, reactive oxygen species, and inflammatory cytokines. The model is supported by several experimental explant culture findings. Interestingly, we found that when post-injury inflammation is mitigated, cartilage composition can partially recover. We suggest that mechanobiological models including cell–tissue-level mechanisms can serve as future tools for evaluating high-risk lesions and developing new intervention strategies.

## 1. Introduction

Joint injuries trigger biological cell signaling pathways implicated in articular cartilage degeneration [1–3]. Cartilage has a limited innate capacity for repair, so when joint injuries cause loss of chondrocyte (cartilage cell) viability and extracellular matrix (ECM) components, it is often irreversible. Ultimately, these sequelae of joint injury lead to post-traumatic osteoarthritis (PTOA), a disease marked by pain in the affected joint [1,2]. The mechanisms of the onset and progression of PTOA are poorly understood, but several intertwined factors have been identified: chondrocyte death [4,5], mitochondrial dysfunction and the subsequent overproduction of reactive oxygen species (ROS) [6,7], inflammation [2], and increased proteolytic activity triggered by excessive mechanical loading [8,9].

Mechanical loading is an important factor in chondrocyte-regulated cartilage homeostasis [10,11]. Injurious loading may initiate ECM degeneration [1,7,12] and cause cell death including apoptosis and necrosis [11,13–16]. This degenerative pathway may be further promoted locally by dynamic loading, even if compressive tissue-level mechanical strains remain within physiological limits [17]. Necrosis is an acute form of cell death caused by direct mechanical damage to cells such as injurious single-impact loading or high local strains and/or strain rates [13,14,18]. Necrosis is also suggested to result in the release of damage-associated molecular patterns (DAMPs) and pro-inflammatory cytokines [19–21] and lead to ECM degeneration caused by proteolytic enzymes [22]. In addition, near the injury site, excessive local strains may alter cell function. For instance, associated changes in mitochondrial activity and physiology can culminate in the excessive production of ROS [23,24]. Apoptosis, the controlled subacute form of cell death, has also been associated with excessive production of ROS [15,25]. Excessive ROS production has been suggested to promote ECM degeneration via decreased matrix biosynthesis [26], increased release of proteolytic enzymes [27,28], and inhibition of tissue inhibitors of metalloproteinases (TIMPs) [26,29].

Inflammation is another important factor in cartilage homeostasis. During the early phases of PTOA, pro-inflammatory cytokines such as interleukin-1 (IL-1), IL-6, IL-18, and tumor necrosis factor-α (TNF-α) originating from the synoviocytes of the synovial lining are secreted to the synovial fluid. These molecules then diffuse and advect into cartilage causing reduced biosynthesis and predisposing the tissue to degeneration that is counter-balanced by anti-inflammatory cytokines (*e.g.*, IL-4, IL-10, IL-13) [30], TIMPs [31], and growth factors (*e.g.*, insulin-like growth factor-1) [2,30,32]. Prolonged inflammation may shift cartilage homeostasis to the catabolic state, in which the ECM is degraded via aggrecanases (*e.g.*, a disintegrin and metalloproteinase with thrombospondin motifs-4,5; ADAMTS-4,5) and collagenases (*e.g.*, matrix metalloproteinases-1,3,13; MMP-1,3,13) [2,30,31].

The ability to predict cartilage degeneration via both biomechanical and inflammatory mechanisms is critical to comprehending disease progression, evaluating the efficacy of medical treatments, and developing new intervention strategies. Previous computational finite element models have introduced promising frameworks to simulate the biomechanics- and inflammation-driven cartilage degeneration at joint, tissue, and cell-levels in both spatial and temporal manner [17,33–36]. Previous biomechanics-driven computational models have targeted the main cartilage injury mechanisms including necrosis, apoptosis, and pro-inflammatory cytokine and DAMP-signaling but without including the subsequent degeneration of different ECM components [35,37,38]. More recently, strain/stress threshold-based modeling approaches have been developed to predict tissue-level proteoglycan (PG) loss without explicitly modeling the underlying chondrocyte-regulated mechanisms [33,39]. Yet, there are no numerical approaches that would model both trauma-related cellular mechanisms and subsequent changes in cartilage composition.

Thus, we aimed to develop a new 2-D cell-and-tissue-level mechanobiological model of cartilage degeneration [17,34,36] to localize and predict injury- and inflammation-related chondrocyte responses and subsequent early-stage cartilage degeneration. We did not model the injurious loading *per se*, but we instead concentrated on how cell death and compositional changes evolve in injured cartilage that is possibly experiencing locally elevated strains post-injury. We hypothesized that i) injury-related cell necrosis and apoptosis and PG loss occur at early time points near lesions while ii) inflammation-mediated PG loss occurs later and in more distant intact areas. To predict tissue-level cell death and PG loss in an injured environment, we simulated three different injury-related cell mechanisms separately and simultaneously. In the numerical model, excessive biomechanical shear strains trigger I) necrosis and II) apoptosis following cell damage (e.g., mitochondrial dysfunction) and ROS overproduction, while IL-1 diffusing into the tissue trigger III) inflammatory responses. We qualitatively compared the simulated cell death and PG content distributions with stained histological cartilage sections from previous *in vitro* experiments [17,40]. To address the lack of quantitative experimental data, we conducted a sensitivity analysis for the most relevant parameters in the model, which were selected based on preliminary simulations (necrosis/cell damage rate, ROS production rate, rate of spontaneous apoptosis, and decay rate of IL-1 concentration). Our approach is a novel step towards modeling PTOA progression through chondrocyte-driven biological mechanisms triggered by both locally excessive biomechanical loading and inflammation.

## 2. Materials and methods

A computational mechanobiological model, inspired by previous models [17,34,36,37], was developed to simulate cartilage degeneration in experimental cartilage geometry after injurious unconfined compression to explain biological tissue-level damage via cell-driven mechanisms

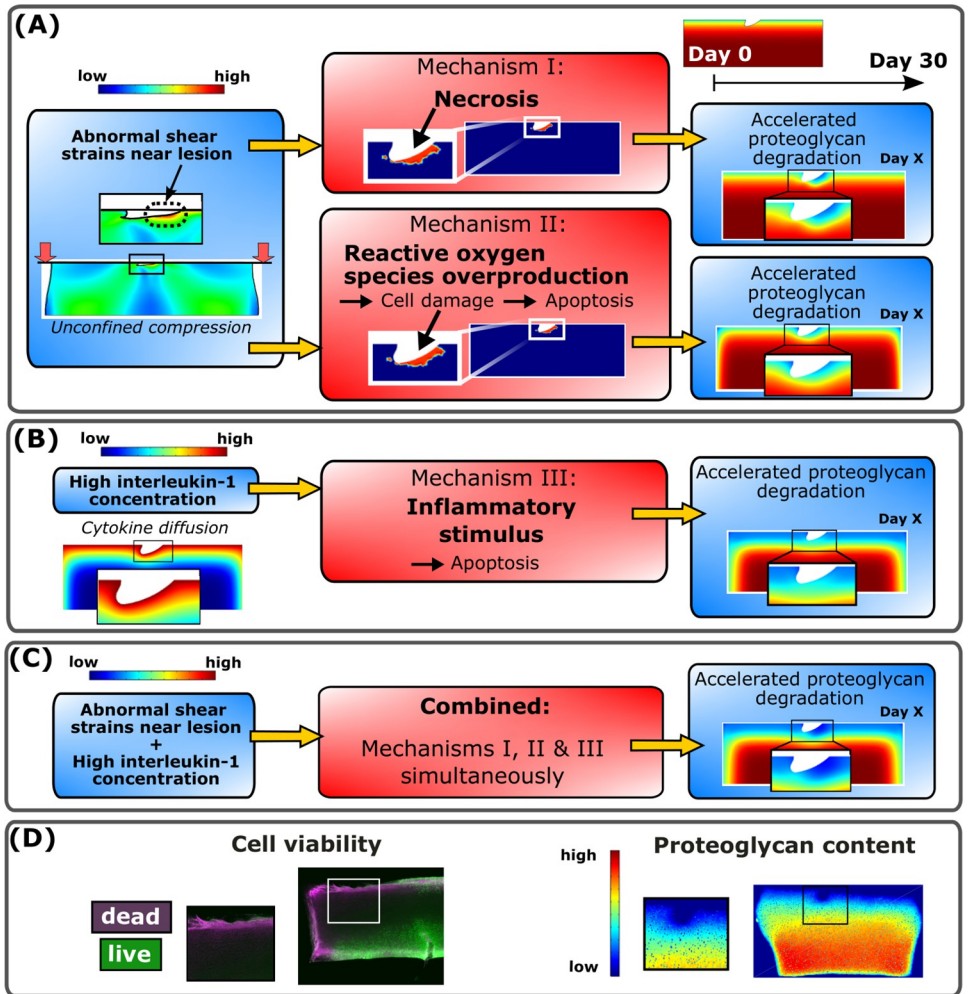

**Fig 1. Computational modeling framework and comparison against biological data.** Delineation of the simulated mechanisms I-III in the proposed computational model to predict temporal and spatial changes in cell viability and proteoglycan (PG) loss over 30 days. (A) Unconfined compression (15% axial strain, 1 Hz loading frequency) of injured cartilage was simulated to obtain maximum shear strain distributions. Two biomechanically-induced degradation mechanisms were triggered locally in regions experiencing abnormal maximum shear strains; chondrocyte necrosis (mechanism I) and chondrocyte damage-associated overproduction of reactive oxygen species (ROS) followed by cell apoptosis (mechanism II). (B) Interleukin-1 (IL-1) diffusion (1 ng/ml of IL-1 in the culture medium) in cartilage caused spatially more distributed inflammatory stimulus which led to chondrocyte apoptosis (mechanism III). Moreover, all the mechanisms I-III accelerated the PG degradation by decreasing the PG biosynthesis and increasing the proteolysis of PGs. (C) Finally, the combined model was developed to simulate the synergistic effects of mechanisms I-III. (D) Simulated cell viability and PG content were also qualitatively compared against experimentally measured cell viability and digital densitometry measurements (an estimate of PG content).

[17,40]. The cartilage PG degeneration was controlled with three different adaptive mechanisms (Fig 1): shear strain-induced I) immediate necrosis of a cell population and II) ROS overproduction and subsequent cell apoptosis (Fig 1A). These injury-related mechanisms ultimately resulted in an increased aggrecanase release. The last mechanism III) is associated with the effects of IL-1 (Fig 1B), which can cause chondrocyte apoptosis as well as upregulation of aggrecanase production in the remaining live cells. All three mechanisms were assumed to lead to decreased PG biosynthesis after decreased cell viability. These mechanisms were modeled separately and simultaneously in a combined model (Fig 1C). We simulated the evolution

of the viable cell and matrix PG content distributions for 12 days, while also providing extrapolated insights up to 30 days. Based on the simulated results, we quantitatively analyzed near-lesion (0.1 mm from lesion edges [17,34]) and bulk (the whole cartilage geometry) cell viability and PG loss at several time points. The simulated results in an injured cartilage explant model were also qualitatively compared with previous explant culture experiments (Fig 1D) [17,40].

## 2.1. Comparative biological data

Predictions of our theoretical computational model were qualitatively compared against histological changes observed in the previous explant culture experiments (Fig 2) [17,40]. We emphasize that the exact experimental protocol was not modeled, thus no quantitative comparison is provided. We find this qualitative comparison feasible since the goal of this study was to gain an understanding of the possible underlying mechanisms to explain experimental findings in PTOA-like conditions.

In the experiments (Fig 2) [17,40], cylindrical articular cartilage explants (diameter 3 mm, thickness 1 mm) were prepared from patellofemoral grooves of freshly slaughtered 1–2-week-

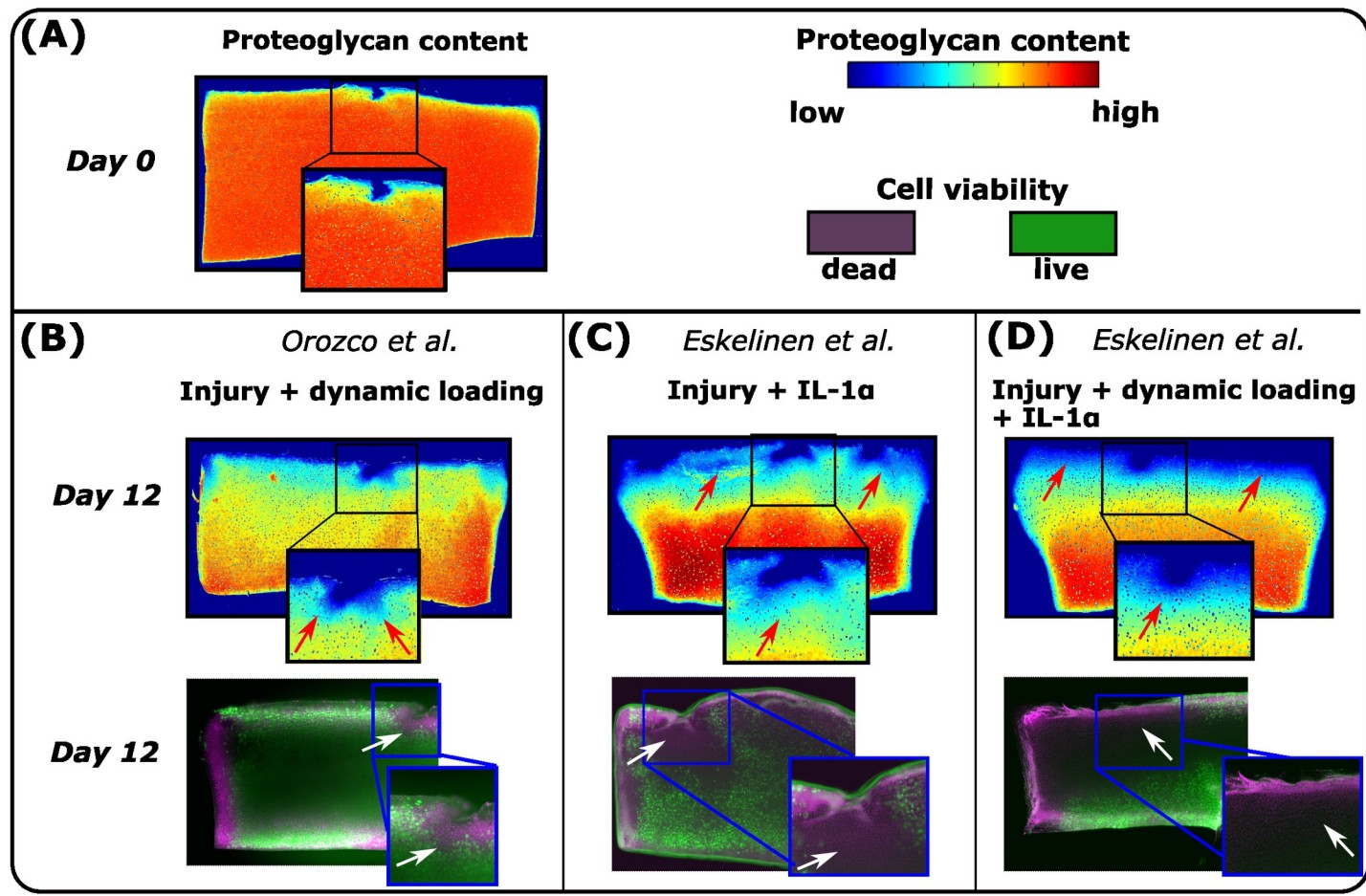

**Fig 2. Previous experiments.** In the previous experiments conducted by Orozco et al. [17] and Eskelinen et al. [40], the injured, dynamically loaded, and IL-1-challenged cartilage samples were analyzed at several time points during 12-day cultures. Cell viability and optical density (an estimate of proteoglycan (PG) content) were measured with fluorescence microscopy and digital densitometry, respectively. (A) At day 0, PG loss in cartilage was minor. At day 12, the results showed (B) substantial cell death and PG loss near lesions after dynamic loading in the injured cartilage. IL-1 challenge induced cell death and PG loss also in the intact areas (C) without and (D) with dynamic loading post-injury. Red arrows highlight locally low optical density and white arrows locally high cell death.

old calves. The explants were subjected to injurious compression (50% strain, 100%/s strain rate) with 1) compressive dynamic loading (15% strain amplitude, 1 Hz haversine waveform, 1 hour loading periods 4 times per day), 2) IL-1-challenge (1 ng/ml), or 3) a combination of dynamic loading and IL-1 challenge. A free-swelling control group was also included for comparison. Cell viability and spatial optical density [41] were assessed at several time points up to 12 days with fluorescence microscopy and digital densitometry.

The experiments [17,40] showed minor PG loss between intact and injured sample regions on the day of injury (Fig 2A, day 0). Qualitatively, cell death was high and PG content low in the injured and dynamically loaded group mostly near lesions (Fig 2B, day 12 vs. day 0, white and red arrows). After injury and IL-1 treatment, PG content was noticeably low near all edges of the cartilage plug (Fig 2C, red arrows). Dynamically loaded injured and inflamed plugs also showed markedly low PG content both away and near lesions (Fig 2D).

## 2.2. Simulation of abnormal biomechanical shear strains promoting necrosis and cell damage

A finite element model of injured cartilage was subjected to physiologically relevant dynamic loading as in a previous study [17]. The cartilage injury (lesion) and simplified dynamic loading protocol (two loading–unloading cycles in unconfined compression instead of continuous hour-long loading) in the simulations were implemented based on the experiments [17]. Importantly, we did not model the injurious loading itself, but rather the subsequent physiologically relevant dynamic loading of injured cartilage. The mechanical behavior of cartilage was modeled using a fibril-reinforced porohyperelastic material with Donnan osmotic swelling [42]. The material model input incorporated depth-dependent material properties including water content, PG content, and collagen orientation and density [17] (see S1 Text and S1 Table). This material model has been shown to reliably capture cartilage mechanical behavior [42,43]. The model output was maximum shear strain distribution, showing locally elevated shear strains near the lesion, even though tissue-level loading remained within physiological limits [17,39] (Fig 1A). The mechanical model was constructed in ABAQUS (v. 2021, Dassault Systèmes, Providence, RI, USA), and solutions were obtained using 'soil consolidation' analysis (transient analysis of fully saturated fluid-filled porous media) with the same model geometry and finite element mesh that was assured to converge in our previous work (918 linear axisymmetric elements with pore pressure, element type: CPE4P) [17]. Boundary conditions were assigned as in the previous model (see S2 Text). Since excessive shear strains have been suggested to trigger necrosis and apoptosis in cartilage [18], we used the maximum shear strain distribution as a driving parameter for the locally triggered cell death and PG loss (see section 2.3 for details). As a preliminary test, we conducted simulations with higher compressive strain amplitude to estimate areas experiencing cell necrosis/damage triggered after dynamic high-strain tissue level compression (40% unconfined axial compressions, 1 Hz loading frequency). For more details related to mechanical modeling, please see Fig A in S3 Text.

## 2.3. Modeling cell death and PG loss

**Diffusion of aggrecanases and decrease in PG biosynthesis.** Injury-related cell death and damage, as well as diffusing inflammatory cytokines, may lead to a release of aggrecanases [8,22]. In our model, mechanisms I–III (Fig 1) regulated the amount of released aggrecanases diffusing in cartilage and suppressed PG biosynthesis after decreased cell viability, both leading to PG loss. Also, PGs may be transported out of the tissue through the cartilage–fluid-interface via diffusion. These mechanisms were modeled with time-dependent reaction–diffusion

partial differential equations [36]

$$\frac{\partial C_i}{\partial t} = D_{e,i} \nabla^2 C_i + R_{i,\text{syn}} - R_{i,\text{deg}}, \tag{1}$$

where $t$ is time, $C_i$ is the concentration of the biochemical species $i$ (aggrecanases, PGs, IL-1, viable/necrotic/damaged cells), $D_{e,i}$ is the effective diffusivity (zero for cell populations, as we assumed no cell migration), $R_{i,\text{syn}}$ is the source (synthesis) term, and $R_{i,\text{deg}}$ is the sink (degeneration) term of the species $i$. The source/sink terms utilized Michaelis–Menten kinetics to model the production and degradation processes (see S4 Text) and binding of IL-1 to its receptor as in the model by Kar et al. [36]. For example, an increase in the aggrecanase concentration increases the PG sink term, whereas cell death decreases the PG source term. The initial PG content was obtained from the previous experiments [36,44]. In the current study, we did not consider fluid flow-induced advective transport of IL-1, aggrecanases, or PG in Eq (1). For more detailed information on the reaction–diffusion model, readers are referred to S4 Text. Diffusion and reaction of species $i$ were modeled in COMSOL Multiphysics (version 5.6, Burlington, MA, USA) using a 2405-element triangular mesh (Fig A in S5 Text).

**Mechanism I. Necrosis.** First, regions presumed to experience early necrosis due to high mechanical strain [14,18,45] were obtained from ABAQUS simulations using a custom-written (see S6 Text) MATLAB script (R2018b, The MathWorks, Inc., Natick, MA, USA). Based on earlier studies, we assumed that when the maximum shear strain in an element exceeded a threshold of 50% [17], 40% of cells were assumed to become necrotic [46]. These live and necrotic cell distributions were then imported into COMSOL.

The presence of necrotic cells was assumed to result in a rapid increase of local aggrecanase concentration. The imported necrotic cell distribution then served as an initial condition for the enzymatic (aggrecanase-induced) PG degradation. Acute necrosis-driven PG degeneration via aggrecanases is supported by experimental findings showing rapid cell death within hours after single impact loading [46], rapid release of aggrecanases near injured cartilage surfaces [47], and necrosis-driven release or stimulation of proteolytic enzyme activity [22]. According to our preliminary tests, this choice also resulted in early cell death and PG loss near cartilage lesions as observed experimentally [17,40]. In addition, it has been suggested that high local strains during repetitive dynamic loading in injured cartilage could lead to accumulated cell death and possibly secondary necrosis in the superficial zone [48,49], promoting the localized release of inflammatory factors [20–22] which could increase the proteolytic activity associated with the surviving cells [30]. Thus, we assumed an acute aggrecanase release (concentration $C_{\text{aga,init}}$) from necrotic cells $C_{\text{n,c}}$ at the beginning of the simulation:

$$C_{\text{aga,init}} = c_{\text{aga,n,c}} C_{\text{n,c}} = c_{\text{aga,n,c}} p_{\text{n,c}} C_{\text{h,c,0}}, \tag{2}$$

where $c_{\text{aga,n,c}}$ is a calibration constant for the released aggrecanase ($1.2 \times 10^{-19}$ mol) based on a visual comparison of simulated PG concentration and histologically observed PG content findings [40], $p_{\text{n,c}} = 0.4 = 40\%$ is the fraction of necrotic cells [46], and $C_{\text{h,c,0}} = 1.5 \times 10^{14} \frac{1}{\text{m}^3}$ is the initial concentration of healthy cells [50].

**Mechanism II. Damaged cells, ROS release, and apoptosis.** Similarly, as with necrosis, we assumed that 40% of the cells experiencing the maximum shear strains > 50% will become 'damaged cells' $C_{\text{d,c}}$ (e.g., experiencing mitochondrial dysfunction) [17]:

$$C_{\text{d,c}} = p_{\text{d,c}} C_{\text{h,c,0}}, \tag{3}$$

where $p_{\text{d,c}} = 0.4 = 40\%$ is the fraction of damaged cells [51]. Based on observations of increased ROS production in response to excessive mechanical loading [15,24,51], we assumed that the

localized ROS concentration $C_{\text{ROS}}$ increases as a function of damaged cell concentration $C_{\text{d,c}}$ [37]:

$$\frac{\partial C_{\text{ROS}}}{\partial t} = s_{\text{ROS}} C_{\text{d,c}} - \delta_{\text{ROS}} C_{\text{ROS}}, \tag{4}$$

where $\delta_{\text{ROS}} = 6.9 \times 10^{-4} \frac{1}{\text{s}}$ is the ROS decay rate [37] and $s_{\text{ROS}}$ is the ROS synthesis rate described as

$$s_{\text{ROS}} = 0.05 \times s_{\text{max}} = 0.05 \times \frac{10 \text{ nmol}}{1\text{h} \cdot 10^6} \approx 1.4 \times 10^{-19} \frac{\text{mol}}{\text{s}}, \tag{5}$$

where $s_{\text{max}}$ is the estimated maximum oxygen consumption rate (5–21% oxygen tension) [37,52]. Moreover, since the ROS production in healthy cartilage has been estimated to be 1–3% of the maximum oxygen consumption [25,37,53], we assumed 5% ROS production in injured cartilage (overproduction). We assumed no diffusion of ROS since the approximate half-life of the mitochondrial ROS is relatively short (< 1 ms) [54].

Excessive ROS production has been suggested to result in apoptosis and PG loss [15,55]. The former phenomenon was incorporated as damaged cells $C_{\text{d,c}}$ turning apoptotic in an exponential manner [56,57]:

$$\frac{\partial C_{\text{d,c}}}{\partial t} = P_{\text{ROS}} = -k_{\text{d,c}} e^{\tau_{\text{ROS}} C_{\text{ROS}}} C_{\text{d,c}}, \tag{6}$$

where $P_{\text{ROS}}$ describes the rate of damaged cells turning apoptotic due to ROS, $k_{\text{d,c}} = 1.3 \times 10^{-6} \frac{1}{\text{s}}$ is cell death rate for damaged cells [58], and $\tau_{\text{ROS}}$ a calibration coefficient for ROS-dependent cell death ($0.7 \times 10^2 \frac{\text{m}^3}{\text{mol}}$).

Furthermore, PG loss was increased due to increased stimulus of aggrecanase production. The stimulus was modeled with a stimulus equation originally introduced by Kar et al. [36] (see S4 Text). Finally, PG degeneration was modeled based on Eq (1).

**Mechanism III. Inflammation-induced apoptosis.** Pro-inflammatory cytokine-mediated apoptosis was implemented with IL-1 in the following exponential equation [59]

$$\frac{\partial C_{\text{l,c}}}{\partial t} = P_{\text{IL}-1} = -k_1 e^{\tau_{\text{IL}-1} C_{\text{IL}-1}} C_{\text{l,c}}, \tag{7}$$

where $C_{\text{l,c}}$ is the concentration of live cells ($C_{\text{l,c}} = C_{\text{h,c,0}}$, if only inflammation is considered or $C_{\text{l,c}} = C_{\text{h,c,0}}(1 - p_{\text{n,c}} - p_{\text{d,c}})$ if also necrosis and cell damage are considered in the cells experiencing over 50% maximum shear strain), $k_1 = 7.5 \times 10^{-8} \frac{1}{s}$ is the rate of spontaneous apoptosis (11% of cells are apoptotic after 17 days under free-swelling conditions without exogenous cytokines) [32], $\tau_{\text{IL}-1} = 5.7 \times 10^7 \frac{\text{m}^3}{\text{mol}}$ is a calibration coefficient for experimentally observed IL-1-induced depth-dependent apoptosis [32], and $C_{\text{IL}-1}$ is IL-1 concentration. The chosen IL-1 concentration was 1 ng/ml, implemented as a Dirichlet boundary condition on all the edges except the bottom of the cartilage geometry [32,36]. Cytokine diffusion led to PG loss after loss of cell viability and upregulation of aggrecanases via IL-1-mediated stimulus which were simulated separately and simultaneously (for more details, see S4 Text).

**Combining injury-related and inflammatory mechanisms.** In the combined model, cell death including injury-related I) necrosis, II) apoptosis via ROS overproduction in the damaged cells, and III) IL-1-induced apoptosis were all considered simultaneously. Here, the live cell concentration was affected as described in Eq (7). The damaged cells could turn apoptotic due to ROS overproduction ($P_{\text{ROS}}$, Eq (6)) and inflammation ($P_{\text{IL}-1}$, Eq (7)).

**Table 1. Parameters for the sensitivity analysis.**

| Parameters | Range | Description | References |
|---|---|---|---|
| $p_{n,c}$ [−] | 0.20, **0.40**, 0.60 | Necrosis fraction (Eq (2)) | [46] |
| $p_{d,c}$ [−] | 0.20, **0.40**, 0.60 | Damaged cell fraction (Eq (3)) | [51] |
| $s_{ROS}$ [mol/s] | $0.01 \times s_{max}$, $\mathbf{0.05 \times s_{max}}$, $0.09 \times s_{max}$ [a] | Reactive oxygen species production rate (Eq (4)) | [25,37,53] |
| $k_1$ [1/s] | 0, **7.5×10⁻⁸**, 15×10⁻⁸ | Rate of spontaneous apoptosis (Eq (7)) | [32] |
| $\mu$ [1/s] | 1.2×10⁻⁶, **0**, 5.8×10⁻⁶ | Decay rate of the interleukin-1 concentration (Eq (8)) | [60] |

Parameters and ranges chosen for the sensitivity analysis. Bolded values indicate reference values.

[a] $s_{max} \approx 2.8 \times 10^{-18} \frac{mol}{s}$ [37]

### 2.4. Sensitivity analysis for the computational model parameters

To address the lack of quantitative experimental data needed to calibrate some model parameters, we conducted a computational sensitivity analysis for the essential parameters affecting cell death and PG loss. Based on our preliminary tests during model development, the chosen parameters were necrosis fraction ($p_{n,c}$), damaged cell fraction ($p_{d,c}$), ROS production rate ($s_{ROS}$, healthy and excessive levels), and rate of spontaneous apoptosis ($k_1$; the IL-1-induced aggrecanase stimulus was turned off to perceive the effect of altered PG biosynthesis due to cell death on PG loss; Table 1).

**Decreased IL-1 concentration.** Previous clinical and pre-clinical studies have suggested that inflammation may play a major role in PTOA progression. There is also evidence suggesting that after acute inflammation, the concentration of the pro-inflammatory cytokines can decrease exponentially *in vivo* [60,61]. Hence, to gain insights into the possible resolution of acute inflammation and tissue recovery, we simulated time-dependent slow and fast exponential decreases of IL-1 concentration in the culture medium as

$$C_{IL-1,b} = C_{IL-1,b,0} e^{-\mu t}, \tag{8}$$

where $C_{IL-1,b,0}$ is the initial boundary concentration of IL-1 (1 ng/ml) and $\mu$ is the decay rate of the IL-1 concentration.

## 3. Results

### 3.1. Necrosis

Simulated necrosis was localized near the cartilage lesion (Fig 1A) after dynamic loading (initial impact-loading was not modeled). At day 5, the computational reference model ($p_{n,c} = 0.4$) predicted that 10.8% of the viable cells would be necrotic and 21.6% of PGs would be cumulatively lost within 0.1 mm from the cartilage lesion compared to day 0 (Figs 3A and 4, red line). The simulated PG content decreased rapidly and locally during the first day, followed by partial recovery for the rest of the simulation. Sensitivity analysis revealed that, at day 5, a smaller number of necrotic cells ($p_{n,c} = 0.2$; Fig 4B, blue line) resulted in an average PG loss of 16.4% while a greater number (Fig 4C, blue line) of necrotic cells ($p_{n,c} = 0.6$; Fig 4B, purple line) resulted in an average PG loss of 26.1% (Fig 4D, purple line).

### 3.2. Damaged cells, ROS release, and apoptosis

Cell damage was also observed near the lesion (Fig 1A). The computational reference model (moderate ROS overproduction) showed cumulative cell apoptosis of 6.5% and PG loss of 21.2% near the lesion at day 5 compared to day 0 (Figs 3B and 5, red line). An 80% decrease in ROS

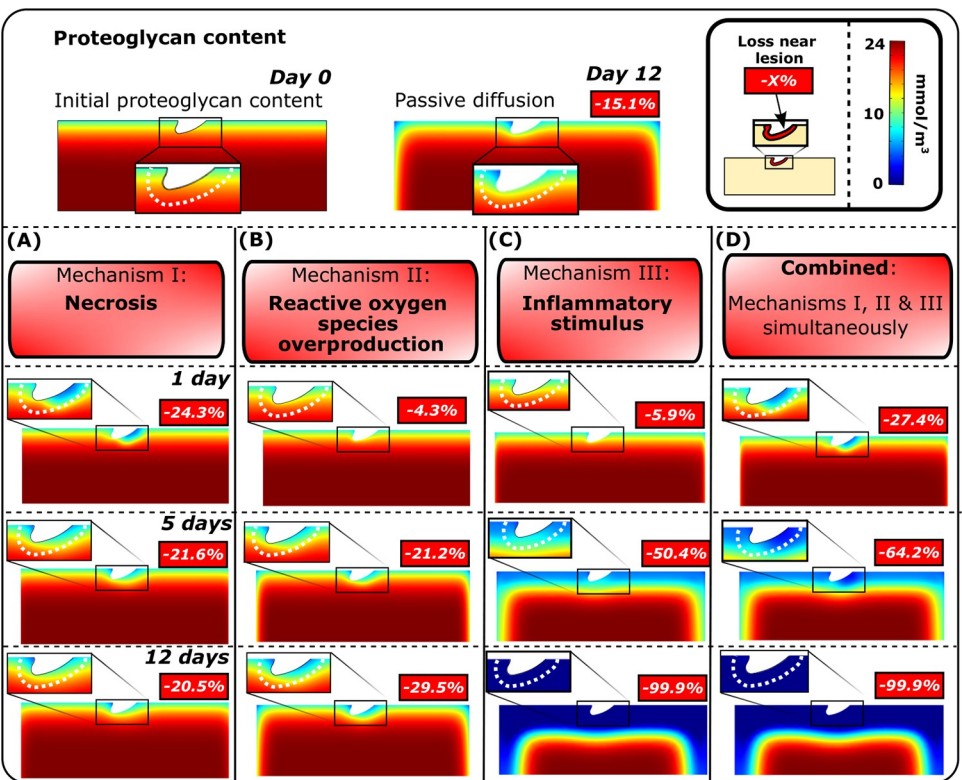

**Fig 3. Simulated proteoglycan degeneration.** Comparison of the simulated spatial changes in proteoglycan (PG) content after A) acute necrosis, B) cell damage, subsequent overproduction of reactive oxygen species and apoptosis, C) inflammatory stimulus, and D) combined mechanisms I, II, and III at days 1, 5, and 12 showed different temporal changes in PG distribution. Percentual changes in the proximity of the simulated lesion (0.1 mm from the lesion edge) were computed relative to the PG content at day 0.

production rate (low, healthy levels; Fig 5, blue line) showed simulated apoptosis of 5.0% and PG loss of 13.0%, whereas increasing ROS production (high ROS overproduction; Fig 5, purple line) to excessive levels led to apoptosis of 7.5% and PG loss of 26.4%. Higher fraction of damaged cells led to higher apoptosis and lower PG content as did the increase of ROS production rate (Fig 6).

### 3.3. Inflammation-induced apoptosis

Diffusion of IL-1 resulted in extensive cell apoptosis and subsequent PG loss near the free surfaces (Figs 3C and 7). The model where PG degeneration via aggrecanases and loss of biosynthesis (induced by apoptosis) was considered, showed PG loss of 50.4% near the cartilage lesion at day 5 (Fig 3C). This rapid degradation masks the effect of IL-1 on PG loss through changes in PG biosynthesis. Thus, in Fig 7, we present sensitivity analysis results with the effect of aggrecanases turned off in the model. At day 5, the reference model ($k_1 = 7.5 \times 10^{-8} \frac{1}{s}$) had PG loss of 11.2% (apoptosis of 33.5%) compared to day 0 (Fig 7, red line). Corresponding models without apoptosis ($k_1 = 0$) exhibited PG loss of 10.2% (Fig 7, blue line; passive PG diffusion) and models with a higher apoptosis rate ($k_1 = 15 \times 10^{-8} \frac{1}{s}$, Fig 7, purple line; apoptosis of 54.2%) showed a PG loss of 11.9% in the cartilage.

### 3.4. Synergistic effect of necrosis, ROS, and inflammation

Cartilage subjected simultaneously to the simulated effect of injury-related and inflammatory mechanisms revealed vast cell death and PG loss near the free surfaces and the lesion (Figs 3D

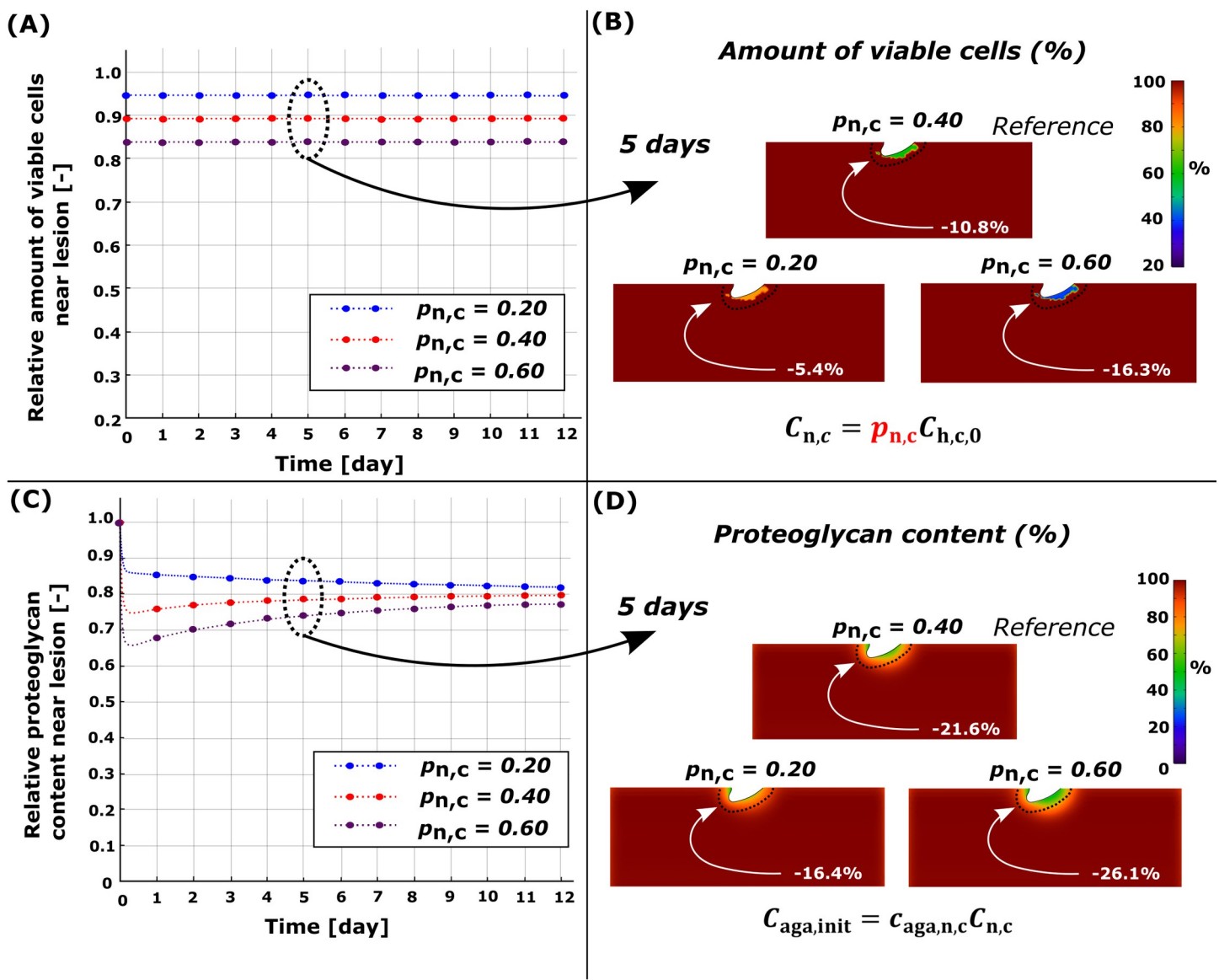

**Fig 4. Sensitivity analysis of simulated necrosis rate $p_{n,c}$.** Comparison of temporal and spatial changes at day 5 (A)-(B) in cell viability and (C)-(D) in proteoglycan (PG) content. (C) Higher necrosis rate led to fast PG degeneration at early time points (days 0–1) and partial recovery of the PG content (days 0–3) near the cartilage lesion. Red line in (A) and (C) refers to the reference model ($p_{n,c} = 0.40$).

and 8A–8D). In the computational reference model (Fig 8A, red line), at day 5, cumulative near-lesion cell death was 46.8% (Fig 8C and 8D, total (bulk) cell death of 11.0% in the whole geometry) and PG loss was 64.2% (total PG loss of 18.9%) compared to day 0 (Fig 8E and 8F, red line). When the IL-1 concentration was decreased slowly in the combined model ($\mu = 1.2 \times 10^{-6}\frac{1}{s}$, Fig 8A, blue line), the simulated near-lesion cell death was 36.3% (Fig 8C and 8D, blue line, total cell death of 8.1%) and PG loss was 62.0% at day 5 (Fig 8E and 8F, blue line, total PG loss of 16.7%). Rapid decrease ($\mu = 5.8 \times 10^{-6}\frac{1}{s}$, Fig 8A, purple line) of IL-1 concentration in the culture medium led to near-lesion cell death of 25.6% (Fig 8C and 8D, total cell death of 5.1% in the whole geometry) and PG loss of 50.8% at day 5 (Fig 8E and 8F, total PG loss of 10.9%). Interestingly, notably less PG loss was observed in 12-day simulations compared to the reference model (Fig 8B).

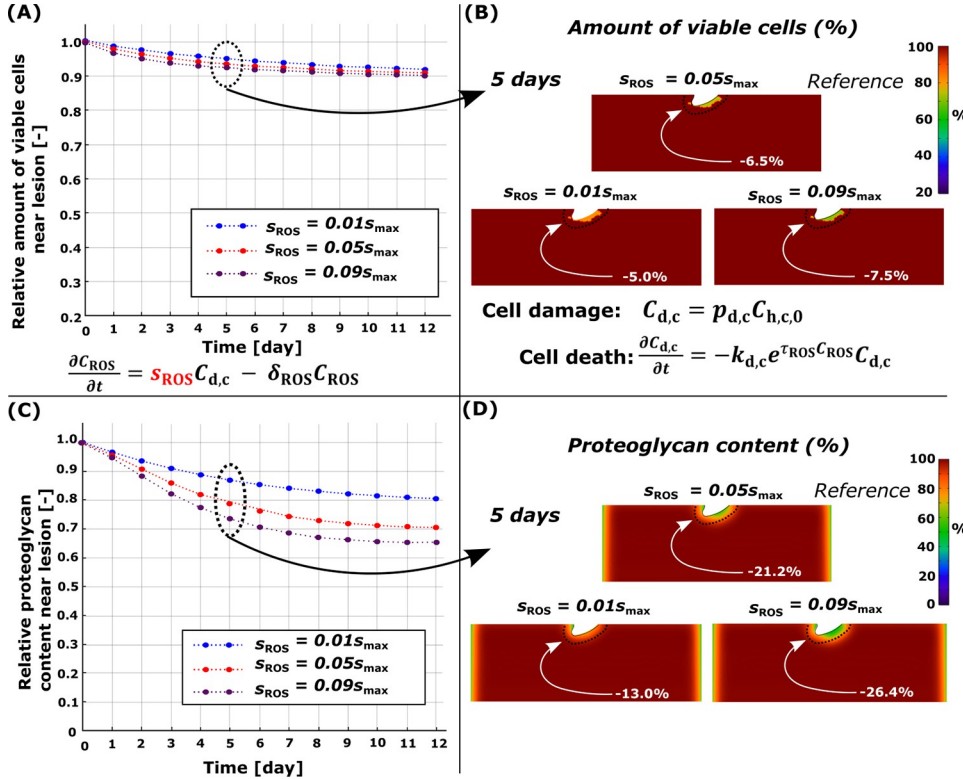

**Fig 5. Sensitivity analysis of simulated reactive oxygen species (ROS) production rate $s_{ROS}$.** Comparison of temporal and spatial changes at day 5 (A)-(B) in cell viability and (C)-(D) in proteoglycan (PG) content. (C) Higher simulated ROS production showed more intensive temporal PG loss and (A) cell death near the cartilage lesion compared to moderate and low production rates. Red line in (A) and (C) refers to the reference model ($s_{ROS} = 0.40$).

**Partial recovery of the PG content in cartilage.** When the simulation was continued until day 30, we observed that the greatest cumulative near-lesion PG loss of 98.5% and 58.2% occurred at days 17 and 9 in the models with slow and fast decrease of IL-1 concentration, respectively. Moreover, we observed that at day 30, the PG content had recovered by 9.4%-points and 20.4%-points around the lesion (corresponding 4.0%-points and 3.9%-points bulk tissue recovery) for the slow and fast decrease of IL-1 concentration when compared to the PG content at days 17 and 9, respectively (Fig 8G).

## 4. Discussion

Previous computational models of early PTOA have not explicitly modeled physical lesions, mechanical loading, and the underlying cell-regulated degradative mechanisms of cartilage. In this study, we bridged this knowledge gap and presented a novel mechanobiological model considering physical cartilage lesion, injury- and loading-related cell death, overproduction of ROS, and diffusion of pro-inflammatory cytokines and their effect on cartilage PG content. We compared the model results against previously measured optical density maps from injured calf cartilage explants and noticed matching predictions of the PG content: extensive and localized near the lesions, but more widely spread when IL-1 was added to the medium. Predicted cell death followed the same pattern of damage localization, observed also *in vitro*. The interesting computational findings are that 1) necrosis alone affects PG content rapidly (0–3 days) in the vicinity of the lesion but its effect almost completely fades away over 5 days,

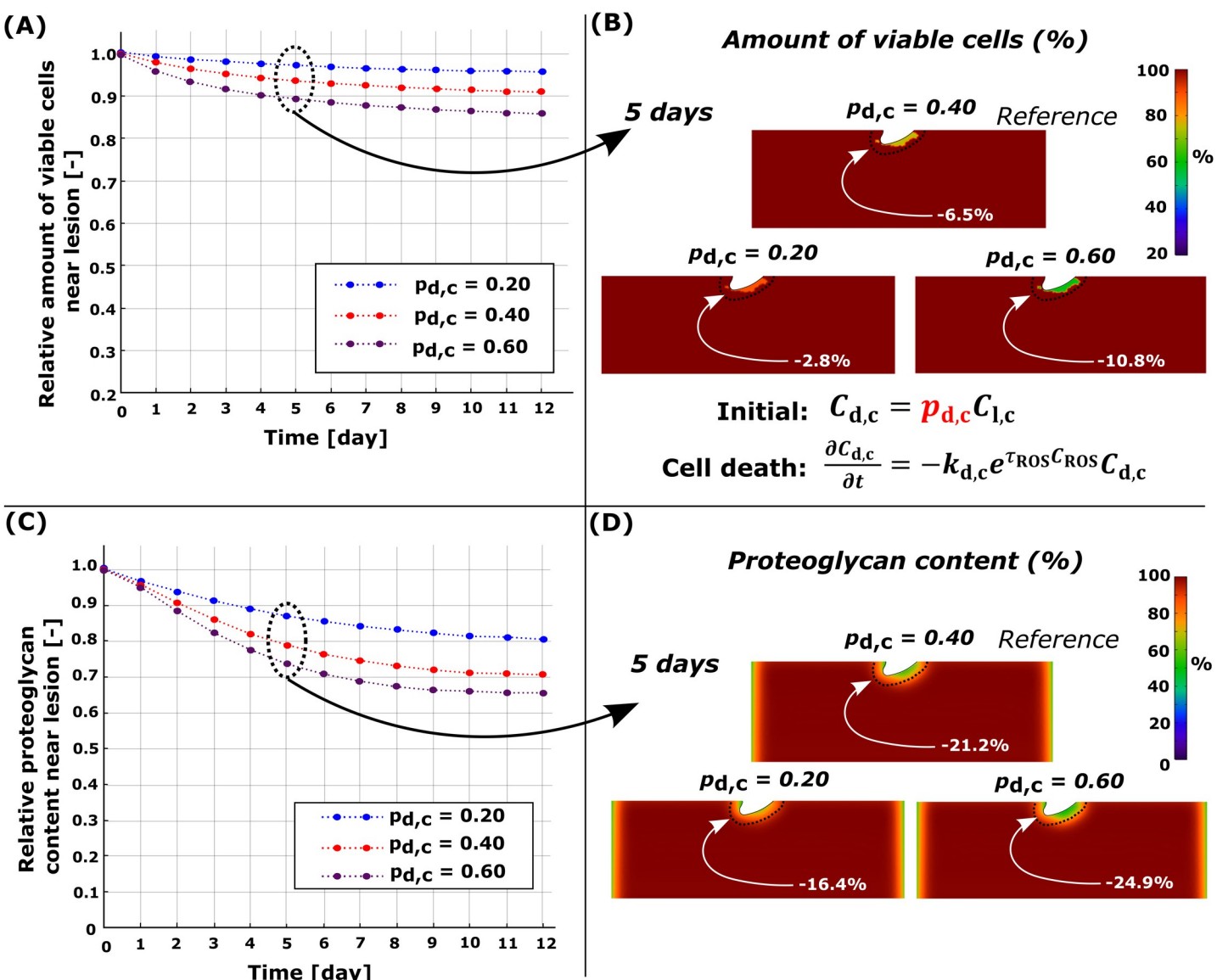

**Fig 6. Sensitivity analysis for simulated damaged cell rate $p_{d,c}$.** Comparison of temporal and spatial changes at day 5 (A)-(B) in cell viability and (C)-(D) in proteoglycan (PG) content. (A) Higher number of damaged cells led to more cell death and (C) more intensive PG degeneration near the cartilage lesion. Red line in (A) and (B) refers to the reference model ($p_{d,c} = 0.40$).

leading to partial recovery of PG content (Figs 3A and 4C), 2) ROS overproduction and especially inflammation have longer-term (over 5 days) effects on PG content (Figs 3, 5C, 6C and 7C), and 3) rapid decrease of IL-1 concentration (leading to lower aggrecanase release and less suppression of PG biosynthesis) facilitates the recovery of PG content even in injured cartilage (Fig 8E). Comparison of the model simulations and biological data implies that our model can pinpoint relevant underlying degradation mechanisms leading to the cartilage degeneration.

## 4.1. Necrosis

Injurious loading may cause rapid (within hours to days) necrotic and apoptotic cell death [13,14,18,55]. The injury can also stimulate catabolic pathways in live cells and predispose them to further harmful mechanical and inflammatory stimuli [62,63]. As a result of high

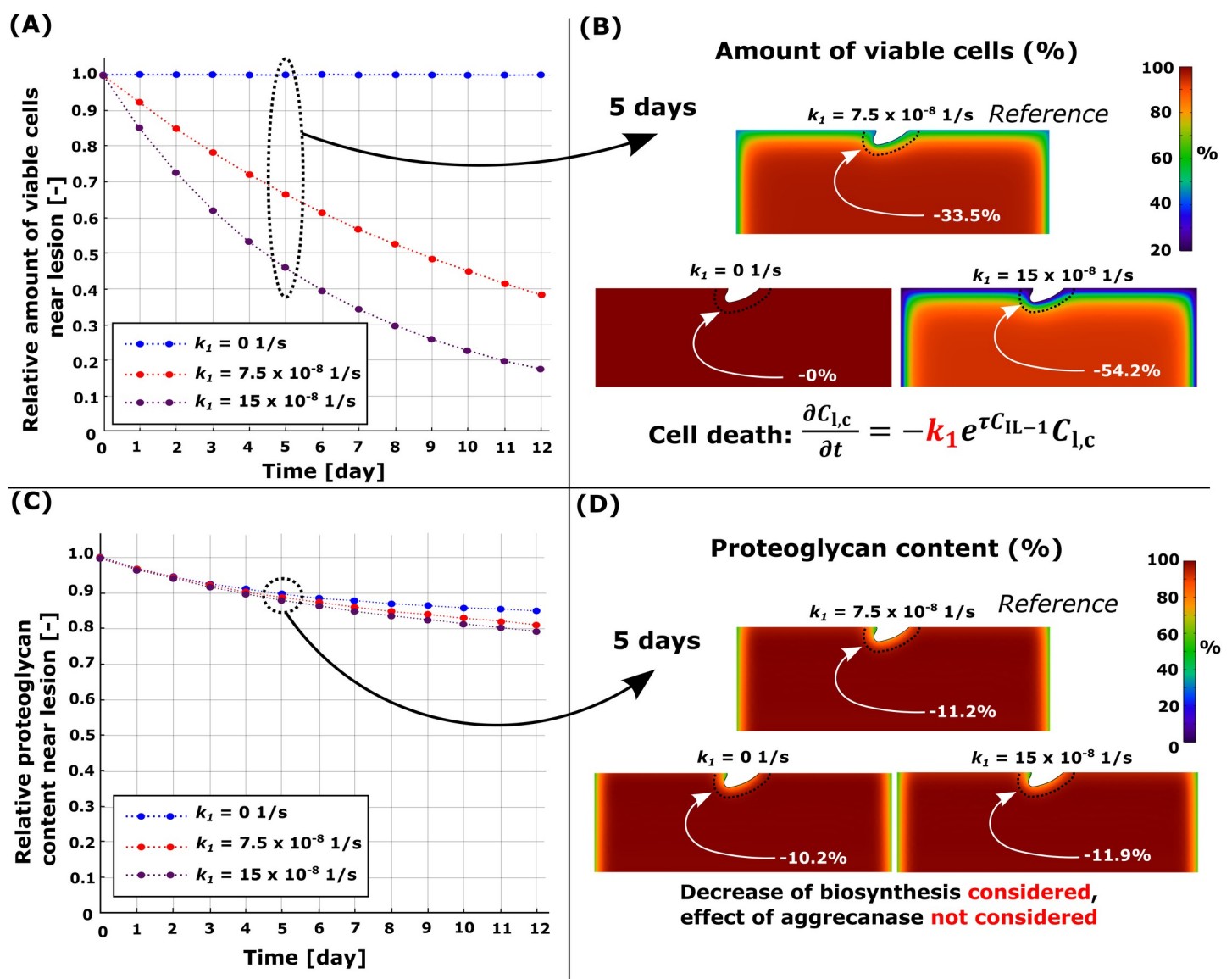

**Fig 7. Sensitivity analysis for the simulated pro-inflammatory cytokine-induced apoptosis rate $k_1$.** Comparison of temporal and spatial changes at day 5 (A)-(B) in cell viability and (C)-(D) in proteoglycan (PG) content. (A) Loss of viable cells and, thus, a decrease of PG biosynthesis (aggrecanase-induced PG degeneration was not considered), had (C) a negligible effect on the simulated PG content over 12 days. Red line in (A) and (B) refers to the reference model ($k_1 = 7.5 \times 10^{-8} \frac{1}{s}$).

susceptibility to harmful stimuli, cells may die or express catabolic enzymes easier than undamaged cells. Locally elevated shear strains near lesions due to mechanical loading could be one such catabolic stimulus, assumed here to lead to localized necrosis and PG loss [17,39]. The cell viability assay with propidium iodide and fluorescein acetate as used by Orozco et al. [17] and Eskelinen et al. [40] does not discern between necrosis and apoptosis, but other studies have shown that similar injurious loading may cause necrotic cell death [46]. Furthermore, the assumption that necrotic cells would release DAMPs inducing inflammatory response (such as IL-1 production, which later causes aggrecanase release [20]) is supported by several studies [21,64]. In addition to necrosis, necroptosis (regulated necrosis) can be involved in the release of DAMPs after injury stimulating the inflammatory response in chondrocytes [65] and, thus, participate in accelerated degradation of cartilage post-injury [66].

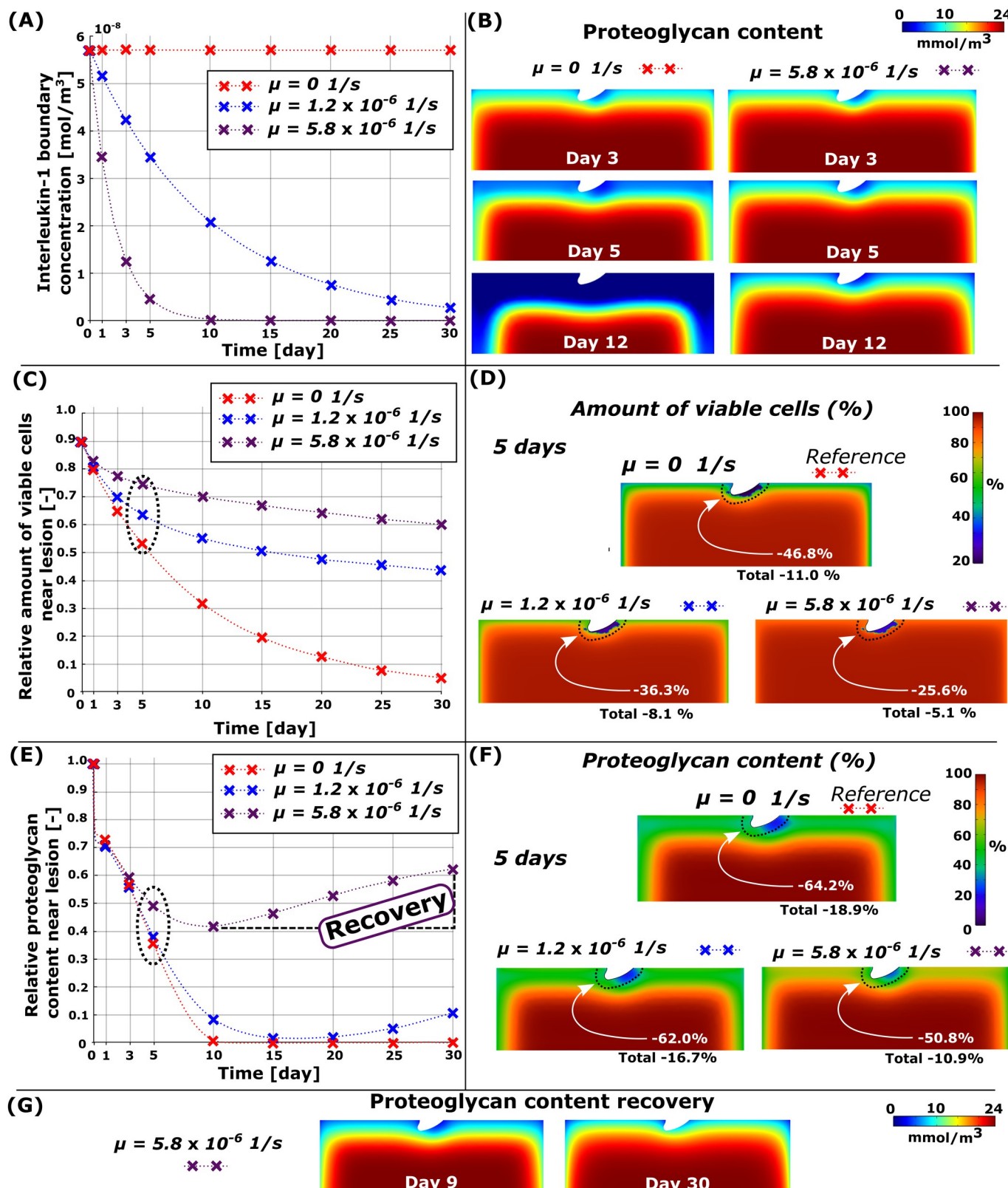

**Fig 8. Simulated decrease of cytokine concentration in the combined model.** (A) Simulated time-dependent exponential decrease of the interleukin-1 concentration in the culture medium and B) comparison of changes in proteoglycan (PG) content with constant ($\mu = 0\ \frac{1}{s}$) and fast-decreasing cytokine

concentration ($\mu = 5.8 \times 10^{-6} \frac{1}{s}$). C) Temporal changes in cell viability in 30-day simulation near the cartilage lesion (within 0.1 mm from the lesion) and D) spatial changes at day 5. D) Temporal changes in the PG content near the cartilage lesion and (F) spatial changes in the whole cartilage geometry (total) at day 5. Decreased exogenous cytokine concentration decreased cell death and PG loss substantially and (G) showed partial recovery of the PG content (here, simulation continued until day 30). Red line in the figure refers to the reference model (decay rate of interleukin-1 concentration $\mu = 0 \frac{1}{s}$).

On average, our model predicted necrotic cell death of 10.8% (40% local necrosis of the viable cells in areas exceeding 50% maximum shear strain threshold, Fig 4A) within 0.10 mm from the lesion at day 5 (compared to day 0) when collagen architecture was based on young bovine cartilage [17,46]. For comparison, Philips et al. [46] reported a high loss of cell viability around the superficial zone of mature bovine cartilage (0.15 mm from the surface), especially in the vicinity of the surface fissures, 1 hour after impact-injury (unconfined compression with ~25 MPa peak stress, 100%/s loading rate). Similarly, human *in vitro* impact-models have also shown necrotic cell death occurring near lesions and increased catabolic expression in cartilage after injury [67,68]. Albeit we did not consider necrosis caused by the initial impact injury, our model could capture the early necrotic cell death near the lesion due to high strains resulting from dynamic loading of injured geometry as well as the following rapid PG degeneration due to high catabolic activity in these areas.

In our simulated necrosis model, aggrecanases were released only at day 0 in response to cell necrosis near the lesion (Fig 4A). We observed the substantial PG loss during day 1 near the lesion (Fig 4C) and, as expected, simulating increased necrosis fraction led to higher PG loss, a scenario that is feasible with high impact loads [12,13,55]. The predicted early burst of enzymatic activity is supported by experiments showing increased aggrecanase expression in injured bovine cartilage 1 day after experimental injury [8]. In addition, Merrild et al. [47] reported localized aggrecanase activity near the lesion 1 day after cartilage injury which could represent newly synthesized aggrecanase or that originating from the tissue reservoirs [9]. Since aggrecanases may be activated intracellularly [69], they may be capable of degrading the ECM PGs when released after cell membrane rupture in necrosis. Moreover, studies about other arthropathies similar to osteoarthritis [22,70] have suggested that the release of aggrecanases occurs in regions experiencing chondrocyte necrosis. Predicted PG degeneration within day 1 and the PG recovery within the following 2 days is explained by rapid outflux of aggrecanases from highly necrotic regions (change of aggrecanase concentration over time is relative to the aggrecanase concentration gradient) and relatively small effect of highly localized necrosis on total PG biosynthesis.

Interestingly, our results from necrosis-only model suggest that cartilage can recover its PG content partially and reach a steady-state in 12 days. This implies that after acute PG degradation and loss, decrease in aggrecanase concentration and diffusion of PGs from deeper layers of the cartilage can promote PG recovery. It has been suggested that early and localized PG degeneration may be part of the repair process of minor cartilage injuries [47], but the continued expression of catabolic enzymes may lead to pathological degradation. However, in previous experiments (Fig 2B) [17,40], PG degeneration was still observed near the lesion at day 12. This suggests that, in addition to the immediate necrosis, further subacute mechanisms associated with cell damage (e.g., oxidative stress and ROS overproduction) should be included in the simulations to better catch the temporal changes in injured cartilage.

## 4.2. Cell damage, ROS, and apoptosis

Since maximum shear strains were excessive near the lesion, the damaged cells producing large amounts of ROS leading to apoptosis were located in the same areas as necrosis. This modeling approach is supported by biological experiments where the amount of ROS was

proportional to the deformation of the chondrocytes [51]. While the simulated necrosis indicated rapid early PG loss followed by partial PG content recovery, damaged cells contributing to the overproduction of ROS led to decreasing PG content over time. This suggests that necrosis might play an early short-term role in PG loss, but cell damage and its downstream catabolic effects may last longer despite the short lifetime of ROS [24,55]. Thus, cell damage and large amounts of ROS could undermine the partial recovery seen with the necrosis-only model and continue cartilage degradation near the lesion even when tissue-level global loading is physiologically normal (15% strain in our model).

Low ROS production in cartilage did not result in major cell death (5.0%), nor did the moderate (6.5%) or severe (7.5%) ROS overproduction (Fig 5B) near the lesion at day 5. Furthermore, low ROS production did not result in a substantial PG loss (13.0%, 2.8% higher than passive PG diffusion) whereas moderate and severe ROS overproduction resulted in higher PG loss, 21.2% and 26.4%, respectively. A similar interplay between damaged cells and increased ROS production leading to cell death and PG loss has been observed experimentally in animal models [24,51,55] and also in human cartilage explants [71].

### 4.3. Inflammation

Simulated inflammation resulted in rapid and substantial cell death and PG loss near the free surfaces, in good agreement with experimental findings (Fig 2C) [32,40]. With 1 ng/ml of IL-1, inflammation-driven degradation mechanisms dwarfed those driven by biomechanics. The inflammation-related PG loss was mostly driven by the aggrecanases; when the proteolytic effect of aggrecanases was turned off, the IL-1-induced apoptosis (resulting in decreased PG biosynthesis) had only a minor effect on the PG loss (Fig 7C and 7D).

Analysis of inflammation-related PG loss has been extensively included in computational and experimental studies [32,34,36]. However, IL-1-induced cell death has rarely been included in computational models. In experimental work conducted by Lopez-Armada et al. [72], ~50% bulk tissue cell death was observed after 7-day culture with 5 ng/ml of IL-1 [72], and Li et al. [32] reported ~50% bulk cell death after 17 days culture with 1 ng/ml of IL-1. Our model exhibited 15.1% and 34.8% bulk cell death on days 7 and 17 with 1 ng/ml, respectively. Lower cell death in our simulated results may be explained by the absence of additional inflammatory mechanisms, such as chondrocyte production of IL-1 after signaling via DAMPs [2,19] or autocrine stimulation via TNF-$\alpha$ [30].

### 4.4. Combined model

Simultaneously acting biomechanical and biochemical mechanisms resulted in marked cell death and PG loss especially near the lesion during the first 5 days (Figs 3D and 8). Later, the IL-1-driven degradation dominated over the other mechanisms around the defect, in agreement with digital densitometry results [40]. Our model was able to capture the well-documented synergistic effect of biomechanics and inflammation on PTOA progression (Fig 2D) [63,73].

Our reference model predicted locally extensive PG loss of 43.6% near the lesion at day 3 (Fig 8B and 8E, red line; total PG loss of 9.0% in the whole cartilage geometry at day 3) and spread of PG loss also to the intact areas at day 5 (Fig 8B and 8F; total PG loss of 18.9%). Previous experiments have shown that injuries can potentiate inflammation causing highly increased expression/activity of matrix-destructing enzymes and PG loss in bovine and human explants [73]. Eskelinen et al. [40] reported increased PG degeneration in the intact regions of injured-and-inflamed bovine cartilage at day 7 compared to day 3. These experiments are in general consistent with our modeling results showing substantial near-lesion PG loss caused

by synergistic effect of inflammation and high shear strains after 3 days and inflammation-induced PG loss also in the intact areas in the subsequent time-points.

Interestingly, a simulated fast decrease of the IL-1 concentration in the culture medium resulted in partial recovery of the near-lesion (20.4%-point increase at day 30 compared to day 9) and bulk PG contents (3.9%-point increase). This finding highlights the major role of inflammation in the computational model; decreasing the cytokine concentration temporally leads to partial recovery of the tissue, while the biomechanical mechanisms contribute to tissue degradation around the lesion. The result of possible partial recovery suggests that inhibition of cytokine activity or rapid cytokine clearance from the culture medium/joint space could suppress production of proteolytic enzymes and rescue PG synthesis via reduced cell death (see Eq (1) and Eq (S14) in S4 Text). Similar PG recovery has been reported in experimental studies [74,75]. Specifically, our numerical model predicted PG recovery between days 10 and 30 (Fig 8E, purple line) after removal of most IL-1 from the culture medium (day 10 in Fig 8A, purple line). During a similar time-window as shown by our model, previous *in vitro* experiments showed partial recovery of uninjured bovine cartilage GAG content over 3 weeks after changing the medium to that without IL-1 [75]. Interestingly, less recovery was observed in the most severely degraded regions compared to the more intact regions of the cartilage after IL-1 treatment. Similarly, our model predicted less recovery in highly degenerated areas near the lesion compared to the more intact areas away from the lesion (Fig 8G). These findings indicate that our model provided reasonable predictions of the cartilage recovery and adaptation.

## 4.5. Limitations

Although our approach represents a step toward elucidating the cell-driven cartilage degradation after injury, our model has limitations that may partly explain the disagreement between the model and experiments.

Biomechanical loading and inflammation of cartilage include many complex cell-level mechanisms. All mechanisms inducing cartilage degradation are nearly impossible to include in a computational model, but we believe cartilage degeneration can be predicted by estimating the net effect of the most significant mechanisms. Here, we have included a few of these mechanisms but, in reality, additional mechanisms may also alter the cartilage degeneration. These mechanisms may include: IL-1 production by chondrocytes [2,30,32], IL-1-induced ROS production [54], ROS-induced necrotic cell death [16,76], introduction of MMP-3-driven matrix degradation after injury [12,77], possible fluid flow-dependent PG loss or cytokine transport through cartilage–fluid-interfaces via advection [17,78,79], and injury-related PG loss due to microdamage and structural changes instead of enzymatic degradation [62]. Moreover, our model does not consider the beneficial effects of moderate dynamic loading, such as alleviation of the catabolic effects of IL-1 and increased synthesis of the ECM matrix components by live cells [40,80,81].

After acute inflammation, physiological IL-1 concentration in the inflamed human knee joint is typically much lower than 1 ng/ml. We simulated diffusion of IL-1 into cartilage (from a bath having 1 ng/ml concentration), the same as used in previous experimental in vitro studies [32,34,36]. Since the biochemical model was calibrated previously based on *in vitro* experiments [36], we did not use physiological concentrations which would just result in slower progress of the degeneration. In the human joint, also other pro-inflammatory cytokines such as TNF-$\alpha$, regulate apoptosis [61,82]. Their effects would be possible to include into the model with Eq (1). However, for now, we did not incorporate TNF-$\alpha$, since we had no information available about its diffusion/reaction coefficients, and it was not used in the experiments utilized for model comparison.

We did not account for the degeneration of the collagen network that would affect the biomechanical properties and cell responses in the cartilage [83]. This was justified as structural and constitutional changes in the collagen network have been observed to occur later than in the PG content [32,84]. Also, we did not explicitly model the pericellular matrix or changes in its properties during the degeneration. There is evidence that alterations in the pericellular matrix properties and cell–matrix interactions may have substantial role in the OA initiation and progression of tissue degeneration [85–87], thus, the pericellular matrix and possibly altered mechanotransduction during loading of damaged cartilage should be accounted for in future studies. In addition, the biomechanical loading used in the computational model is a simplification of the experiments. For instance, we did not simulate the initial impact-loading leading to cartilage defects in the superficial zone or the full dynamic loading protocol used in previous experiments after the injury [17,40]. After impact-loading and during the continuous dynamic loading, changes in PG (and collagen) content can also influence the shear strain distributions [88]. Moreover, considering fluid exchange with the culture medium [89], cartilage injuries [90], and decreased boundary resistance post-injury [91] may induce decreased pressure. This may also broaden the high-strain area from the vicinity of lesions to the deeper layers of the cartilage. Hence, due to simplifications in the biomechanical simulations, our model may underestimate the depth-dependent necrosis, apoptosis, and cartilage degeneration.

Although some inflammation and material parameters have been calibrated well [17,36], full model validation is hampered by the small amount of time-dependent biomechanical and biochemical experimental data from different studies. Data for extensive validation should also include the fraction of cell death (necrosis and apoptosis), ROS activity, and PG content to further improve the calibration of the related parameters and validate the localized degradation processes induced by mechanisms I-III. However, the presented predictions are already generally in line with the current literature and despite the lack of extensive calibration, the current modeling framework can offer insights into the local mechanisms driving cell death and PG loss in PTOA-like conditions.

## 4.6. Future directions

In the future, multiscale mechanobiological models may be feasible to produce patient-specific predictions of early cartilage degeneration. Future developments of the model could include the implementation of the pericellular matrix to investigate the mechano-inflammatory cross-talk between the pericellular matrix and the chondrocyte at the early phases of cartilage injury [87,92]. In addition, extensive experimental research is still needed to elucidate the injury-related mechanotransduction pathways, cell death, and ROS kinetics, which could provide time-dependent quantitative data to calibrate and enhance the current modeling framework. Thus, we are planning to conduct additional experiments to validate our tissue-level model predictions which could then provide validated improvements in the current joint-level models [33,93]. One potential example could be merging this framework with musculoskeletal–finite element modeling workflow [33,94,95] which considers patient-specific joint geometry and loading. When adding inflammatory biomarkers from the synovial fluid, and possibly the defect size, to the workflow with our new cell-level mechanobiological model, virtual evaluation of the effects of biomechanical (such as rehabilitation) and biochemical (for example pharmaceutical treatment) interventions on cartilage health would become possible. This would enable optimizing the treatment and help in clinical decision making. Our current model already considers ROS (Eqs (3)–(6)) and could provide means to simulate, for example, the effect of N-acetylcysteine treatment on oxidative stress, ROS production, cell death, and

cartilage health [7]. Eventually, the model predictions could produce cost-efficient optimized intervention strategies, to mitigate early cartilage degeneration.

## 5. Conclusions

Cell death and enzymatic cartilage degeneration in response to injurious loading are important factors to consider in computational models for predicting PTOA progression. We incorporated biological cell–tissue-level responses including necrotic and apoptotic cell death, ROS overproduction, and inflammation of injured cartilage into a finite element model of early-stage PTOA. Our novel mechanobiological model was able to predict localized cell death and PG loss similar to previous biological experiments; biomechanically induced necrosis and apoptosis and the following enzymatic degeneration of PGs were observed near the cartilage lesion, while diffusing pro-inflammatory cytokines resulted in more widely spread damage. Based on the computational model predictions, rapid inhibition or clearance of pro-inflammatory cytokines would result in partial recovery of the PG content and could be a potential way to decelerate PTOA progression even in injured tissue. In the future, the current computational framework could enhance previous models by introducing new mechanisms, thus providing a better understanding of PTOA progression. Furthermore, in the future, thoroughly calibrated multi-level mechanobiological models could be a valuable tool in assessing patient-specific pharmacological treatments time-dependently and help in the planning of new intervention strategies.

## Supporting information

**S1 Text.** More detailed explanation of the biomechanical material model.
(DOCX)

**S2 Text.** More detailed explanation of the simulations, boundary conditions and finite element mesh of the biomechanical model.
(DOCX)

**S3 Text.** Biomechanical simulations with higher axial strain amplitude to investigate the initial injury.
(DOCX)

**S4 Text.** More detailed explanation of the modeling of the proteoglycan loss with the reaction–diffusion equations.
(DOCX)

**S5 Text.** Mesh sensitivity analysis for the reaction–diffusion model.
(DOCX)

**S6 Text.** More detailed explanation of the data interpolation from Abaqus to Comsol.
(DOCX)

**S1 Table.** Variables describing cartilage composition used in the biomechanical material model.
(DOCX)

## Acknowledgments

We acknowledge the support of University of Eastern Finland, Lund University, Massachusetts Institute of Technology, and University of Iowa to conduct this research.

## Author Contributions

**Conceptualization:** Joonas P. Kosonen, Atte S. A. Eskelinen, Gustavo A. Orozco, Petteri Nieminen, Donald D. Anderson, Alan J. Grodzinsky, Rami K. Korhonen, Petri Tanska.

**Data curation:** Joonas P. Kosonen, Atte S. A. Eskelinen, Gustavo A. Orozco.

**Formal analysis:** Joonas P. Kosonen.

**Funding acquisition:** Rami K. Korhonen, Petri Tanska.

**Investigation:** Joonas P. Kosonen, Atte S. A. Eskelinen, Gustavo A. Orozco, Rami K. Korhonen, Petri Tanska.

**Methodology:** Joonas P. Kosonen, Atte S. A. Eskelinen, Gustavo A. Orozco, Petteri Nieminen, Donald D. Anderson, Alan J. Grodzinsky, Rami K. Korhonen, Petri Tanska.

**Project administration:** Rami K. Korhonen, Petri Tanska.

**Resources:** Rami K. Korhonen, Petri Tanska.

**Software:** Joonas P. Kosonen, Atte S. A. Eskelinen, Gustavo A. Orozco.

**Supervision:** Petteri Nieminen, Donald D. Anderson, Alan J. Grodzinsky, Rami K. Korhonen, Petri Tanska.

**Validation:** Joonas P. Kosonen, Atte S. A. Eskelinen.

**Visualization:** Joonas P. Kosonen.

**Writing – original draft:** Joonas P. Kosonen.

**Writing – review & editing:** Atte S. A. Eskelinen, Gustavo A. Orozco, Petteri Nieminen, Donald D. Anderson, Alan J. Grodzinsky, Rami K. Korhonen, Petri Tanska.

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
