## [Decision Letter · Decision Letter 0]

14 Aug 2022

Dear Mr Kosonen,

Thank you very much for submitting your manuscript "Injury-related cell death and proteoglycan loss in articular cartilage: Numerical model combining necrosis, reactive oxygen species, and inflammatory cytokines" for consideration at PLOS Computational Biology.

As with all papers reviewed by the journal, your manuscript was reviewed by members of the editorial board and by several independent reviewers. In light of the reviews (below this email), we would like to invite the resubmission of a significantly-revised version that takes into account the reviewers' comments.

We cannot make any decision about publication until we have seen the revised manuscript and your response to the reviewers' comments. Your revised manuscript is also likely to be sent to reviewers for further evaluation.

Sincerely,

Inna Lavrik

Associate Editor

PLOS Computational Biology

Jason Haugh

Deputy Editor

PLOS Computational Biology

Reviewer's Responses to Questions

**Comments to the Authors:**

Reviewer #1: In this paper, a numerical approach was developed to study the early post-traumatic biomechanical and inflammatory effects on cartilage homeostasis. The results show that cartilage composition can partially recover when post-injury inflammation is mitigated. This is an interesting study, and the paper is well written. The paper can be accepted after some revisions.

(1) Equation (1) shows the reactive transport process of the biochemical species, which normally involves diffusion governed by diffusion coefficient of the species and advection controlled by interstitial fluid flow in cartilage tissue. It is unclear how advection transport was modelled. In addition, the production, binding and degradation process in cartilage is also important. The authors should provide more details by referring to relevant studies in this field (e.g., https://dx.doi.org/10.3970/mcb.2008.005.133).

(2) For necrosis analysis, the current approach is oversimplified. It has been known that cartilage is a fully saturated porous media, composed of interstitial fluid and solid phase (https://doi.org/10.1016/j.triboint.2020.106185), and the consolidation response of cartilage could be significantly changed there is a defect, and ultimately further damage progression (e.g., https://doi.org/10.1016/j.jmbbm.2017.05.018). The authors should conduct further literature review and provide some in-depth discussions.

(3) The authors stated that the numerical predications are supported by several previous experimental findings. However, the relevant validation process is missing.

(4) In vivo, how is IL-1 transported into cartilage and produced by chondrocytes? Please discuss.

Reviewer #2: In their manuscript, Kosonen et al. report interesting findings of a new combined numerical model to simulate PTOA-associated pathomechanisms, such as cell death and proteoglycan loss in cartilage. Overall, this seems to be a well-planned project and reliable model.

However, there are some questions and comments, which need to be addressed before considering publication.

1 - In comparison to highly hypocellular human cartilage, bovine or porcine cartilage is rather cell-rich. This, of course, also influences the biomechanical properties of the tissue. The numerical model seems to be mainly based on preliminary results from a bovine injury model. In the literature, there are also various reports focusing on the cellular response (cell death and cartilage degeneration) after ex vivo cartilage trauma using human cartilage. The authors should refer to human models and discuss the similarities or differences.

2 - The advantage of this numerical model remains unclear. It seems to “predict” or rather reproduce the findings of preliminary ex vivo cartilage injury models. What exactly is the benefit? Might it be possible to replace in vitro or even in vivo testing? Although, an ex vivo cartilage injury model is already a simplified model: Is it really possible to break down this complex system, including cell-matrix interaction, different cell types (there is not only one type of chondrocytes!), release of various cytokines, chemokines, catabolic enzymes and so on, to a few parameters to predict a potential outcome?

Results part:

3 - In generally, the impact energy and type (repetitive/ cyclic or single impact) should be described in more detail in the results. For example, it is mentioned that “… we did not model the injurious loading itself, but rather the subsequent physiologically relevant dynamic loading of injured cartilage”. The reader should understand the results without searching the missing information in the Materials part.

4 - The authors simulated cell death (apoptosis/ necrosis) at day 5. For example, line 335, “… average cell apoptosis of 6.5% …”

What exactly does the 6.5 % represent? Apoptosis starts after injury and continues until day 5. Does the 6.5 % refers to the cumulative number of death cells or the apoptosis rate at day 5 (without considering the cells, which underwent apoptosis before)?

5 – Why did the authors focus on IL-1b as pro-apoptotic cytokine? TNF is usually the predominant pro-apoptotic/ pro-necroptotic cytokine. Is it because TNF is rather released by synovial cells than chondrocytes? Speaking of which, the authors have not included synovial inflammation but only chondrocyte-derived cytokine. If this is correct, the authors should discuss why they did not include TNF/ synovial inflammation, because this would represent the in vivo situation.

Discussion:

6 - Line 415: “The injury can also sensitize live cells to turn catabolic more easily by later mechanical and inflammatory signals causing more extensive cell death if catabolic signals are not ceased.” This sentence is very complex and hard to understand. Rephrasing is required.

7 - Line 418 ff: besides apoptosis, the authors discuss necrotic cell death after cartilage injury. There is also a report about necroptosis (regulated necrosis) after blunt mechanical impact, which describes the subsequent release of DAMPs, triggering pro-inflammatory processes.

8 - Line 433: “In our simulated necrosis model, aggrecanases were released only at day 0 in response to cell necrosis near the lesion (Fig. 4A).” Could the authors please clarify this statement? Why should aggrecanases be release in response to necrosis (at day 0)? It has previously been demonstrated that expression of some aggrecanases and MMPs is only slightly induced or even downregulated during the acute phase (0-24 h) and peaks a few days after ex vivo trauma (3 – 7 days). This might be linked to oxidative stress and subsequent induction of redox-sensitive pathways (p38, NFkB).

9 - Line 499: The “partial recovery” observed after decrease of IL-1b should be discussed with/ compared to the literature. Why is there an increase of anabolism, which cells contribute to the regeneration of the tissue, … is this recovery induced by the “subsequent physiologically relevant dynamic loading of injured cartilage”? The influence on physiological dynamic loading on cartilage metabolism, in particular anabolism, should be included in the discussion.

10 – Line 543 ff: The future outlook to “produce patient-specific predictions of early cartilage degeneration” is not clear. How exactly should that work? And what exactly does “patient-specific” mean in this context? Which parameter would be necessary to make predictions (e.g. synovial inflammation, defect size, weight, …)??

**Have the authors made all data and (if applicable) computational code underlying the findings in their manuscript fully available?**

Reviewer #1: Yes

Reviewer #2: Yes

PLOS authors have the option to publish the peer review history of their article (what does this mean?). If published, this will include your full peer review and any attached files.

Reviewer #1: No

Reviewer #2: No
---

## [Editor Report · Decision Letter 1]

6 Dec 2022

Dear Mr Kosonen,

We are pleased to inform you that your manuscript 'Injury-related cell death and proteoglycan loss in articular cartilage: Numerical model combining necrosis, reactive oxygen species, and inflammatory cytokines' has been provisionally accepted for publication in PLOS Computational Biology.

Best regards,

Inna Lavrik

Academic Editor

PLOS Computational Biology

Jason Haugh

Section Editor

PLOS Computational Biology

---

## [Editor Report · Acceptance letter]

4 Jan 2023

PCOMPBIOL-D-22-00983R1 

Injury-related cell death and proteoglycan loss in articular cartilage: Numerical model combining necrosis, reactive oxygen species, and inflammatory cytokines

Dear Dr Kosonen,

I am pleased to inform you that your manuscript has been formally accepted for publication in PLOS Computational Biology. Your manuscript is now with our production department and you will be notified of the publication date in due course.

With kind regards,

Anita Estes
